# Changes in limiting factors for forager population dynamics in Europe across the last glacial-interglacial transition

Alejandro Ordonez [1,2,3] ✉ & Felix Riede[1,4]

Population dynamics set the framework for human genetic and cultural evolution. For foragers, demographic and environmental changes correlate strongly, although the causal relations between different environmental variables and human responses through time and space likely varied. Building on the notion of limiting factors, namely that at any one time, the scarcest resource caps population size, we present a statistical approach to identify the dominant climatic constraints for hunter-gatherer population densities and then hindcast their changing dynamics in Europe for the period between 21,000 to 8000 years ago. Limiting factors shifted from temperature-related variables (effective temperature) during the Pleistocene to a regional mosaic of limiting factors in the Holocene dominated by temperature seasonality and annual precipitation. This spatiotemporal variation suggests that hunter-gatherers needed to overcome very different adaptive challenges in different parts of Europe and that these challenges varied over time. The signatures of these changing adaptations may be visible archaeologically. In addition, the spatial disaggregation of limiting factors from the Pleistocene to the Holocene coincided with and may partly explain the diversification of the cultural geography at this time.

As the link between exogenous environmental factors and organismal physiology, demography is vital for understanding evolution, including cultural evolution[1]. The relevance of past demography for understanding culture change in prehistory has long been recognised[2,3]. Demographic conditions impinge on cultural transmission[4–6] but are also clearly implicated in the boom-and-bust patterns of population fluctuations—including periodic extirpations—suggested to have characterised the demographic histories of prehistoric foragers and incipient farmers in many regions[7–10]. Numerous recent studies have focused on the drivers of population expansion to explain the pattern and timing of human colonisation using a variety of ecological comparative approaches[11,12] (but see refs. 13, 14, for a discussion of points of concern of such approaches). As foragers have a high intrinsic growth rate, however, population increase is the default demographic

trajectory in the absence of cultural or environmental constraints. Yet, past populations did not grow substantially, making it particularly germane to understand the factors that curtailed population growth[15,16]. The approach adopted here builds on the central theorem that population sizes would, at any one time, be regulated by the scarcest resource: the limiting factor[17].

Foragers of the recent past persisted in various environments, from the frigid Arctic to tropical rainforests. Each environment offered particular opportunities but also posed specific challenges. Cultural practices and technology provided a means of buffering against or even overcoming some of these challenges. Nevertheless, environmental factors have also been shown to directly, albeit broadly, constrain human palaeodemography. Several earlier studies have pointed to temperature or seasonality as crucial drivers of forager demography

[1]Center for Biodiversity Dynamics in a Changing World, Aarhus University, Aarhus, Denmark. [2]Department of Biology, Aarhus University, Aarhus, Denmark. [3]Center for Sustainable Landscapes under Global Change, Aarhus University, Aarhus, Denmark. [4]Department of Archaeology and Heritage Studies, Aarhus University, Aarhus, Denmark. ✉e-mail: alejandro.ordonez@bio.au.dk

at global or continental scales[18,19]. However, the specific factors that would have capped or even depressed population size are likely to have varied in space and time. Only by understanding these limiting factors can we begin to conduct targeted investigations of how specific forager populations may have overcome them via population-specific genetic adaptations, behavioural modifications or the 'extra-somatic adaptions'[20] that are so characteristic of human culture.

This study focused on forager palaeodemography in Europe from the Last Glacial Maximum (Greenland Stadial 2, GS2) to 8000 years before present (kyBP), a climatically volatile period also known as the Last Glacial-Interglacial Transition[21]. Previous studies have identified population growth and expansion patterns using various methods commonly used in ecological analyses[12,22–24]. Correlations between temperature and overall population density have been identified, suggesting overall increases in energy availability as the key driver of the increase in human population size following the end of GS2[18]. However, regional population collapses have also been suggested to have occurred asynchronously and in different places[9,25]. These regional patterns raise the question of which specific limiting factors acted on forager populations and how these limiting factors varied over space and time.

Like many related studies, we begin with the global ethnographic hunter-gatherer dataset originally assembled by Binford and now digitally available[26,27]. We couple this to a suite of quantile generalised additive models (qGAMs) that describe changes in the maximum (90th percentile), mean (50th percentile), and minimum (10th percentile) population density as a univariate function of environmental variables related to the effect of energy or water availability, productivity, and annual limits and variability (Table 1). A strong correlation between predictor variables is not a major obstacle in our analyses[28] for two reasons. Firstly, our aim is not to determine the best variables to predict population density. Second, we do not compare the relative

effects of evaluated predictors nor establish the variable contributing the most to the deviations from the regional population density average.

Here, instead of simultaneously evaluating groups of variables, as traditionally done in regression analyses using Binford's ethnographic data[11,12,22,29], our analyses focus on defining the possible main driver (i.e. the individual variable) that acted as a limiting constraint (as suggested by refs. 30, 31). Specifically, we generated 1000 different models for each predictor-quantile computation using 70% of Binford's data. Observations to build each model were selected using an h-block cross-validation approach (ref. 32; see Methods). It is essential to highlight that these models seek not to predict past population densities across regions but to reveal the limiting effects of climate on this vital variable as determined by comparing the absolute effects of the evaluated variables. In this way, our analyses serve as a prerequisite for identifying the specific role of human cultural practices and innovations to buffer and even overcome such limiting factors.

Based on this analysis, we hindcast hunter-gatherer population densities between GS2 and 8kyBP in 500-year intervals using the downscaled centennial average conditions of each predictor derived from a transient climatic simulation (SynTrace-21[33]). We only predict population densities for those climatic variables associated with the best-performing univariate qGAM models where performance was defined using a cross-validation approach (see Methods). Population density for each cell within each 500-years interval is then determined as the minimum predicted density for that cell. Population size for each period was determined by summing the products between population size and the area of each grid over the ice-free areas. We define the limiting environmental factor as the variable predicting the lowest mean population density at a given place and time and explore the dynamic changes in these factors as a function of the quantile used to make our predictions. This approach allows us to query the spatial

**Table 1 | Variables used to generate ethnographic-based models of the effect of climate on hunter-gatherer population density and summary of cross-validated deviance explained and predictive accuracy for the evaluated variables**

| Variable name | Acronym | Units | How does the variable determine population density? | Deviance explained Mean [95% CI] | Training set prediction (Pearson correlations)[2] Mean [95% CI] |
|---|---|---|---|---|---|
| Effective temperature * | ET | C | Energy availability | 0.454 [0.377 0.531] | 0.496 [0.473 0.519] |
| Potential evapotranspiration** | PET | mm/yr | Energy availability | 0.33 [0.237 0.422] | 0.376 [0.327 0.426] |
| Net primary productivity *** | NPP | gCarbon*m$^{-2}$*yr$^{-1}$ | Net primary productivity | 0.417 [0.332 0.503] | 0.597 [0.584 0.61] |
| Mean temperature of the coldest month | MCM | C | Annual limit | 0.499 [0.427 0.571] | 0.457 [0.439 0.476] |
| Mean temperature of the warmest month [a] | MWM | C | Annual limit | 0.394 [0.313 0.475] | 0.407 [0.384 0.429] |
| Temperature seasonality (Standard deviation of monthly means) | TS | C | Annual variability | 0.496 [0.42 0.572] | 0.515 [0.487 0.542] |
| Annual precipitation (Log transformed) | TAP | Log$_{10}$ mm/yr | Water availability | 0.336 [0.246 0.425] | 0.587 [0.556 0.619] |
| Precipitation of the driest month (Log transformed) | PDM | Log$_{10}$ mm/month | Annual limit | 0.107 [0.041 0.173] | 0.23 [0.169 0.292] |
| Precipitation of the wettest month (Log transformed) | PWM | Log$_{10}$ mm/month | Annual limit | 0.352 [0.273 0.432] | 0.639 [0.631 0.648] |
| Precipitation seasonality (Coefficient of variation of monthly totals) | PSeson | No units | Annual variability | 0.067 [0.007 0.127] | 0.176 [0.114 0.237] |

[a]Removed due to large non-analogy
*Calculated following ref. 26.
**Calculated following on ref. 114.
***Calculated following ref. 115.
Estimates correspond to those of a 1000-fold cross-validation approach (1000 samples of 70% training and 30% testing observations). The difference between deviance explained and Pearson correlations can be attributed to what is measured in each of these. The deviance explained in the context of a gaussian GAM represents the same as an unadjusted R[2] on a linear regression context[96,97]; hence representing the model improvement from an intercept-only model. The square of the correlation between the training set predicted vs observed values provide a metric of model transferability by indicating the capacity of the models to predict an independent dataset.

dynamics of forager limiting factors across the Last Glacial-Interglacial Transition and derive specific hypotheses as to which selection pressures acted most strongly on different forager communities in Late Pleistocene and Early Holocene Europe.

Our analysis demonstrates that the limiting factors for forager population densities showed marked differences in space and time. Temperature-related variables were the main limiting factors during the Pleistocene, whereas a regional mosaic of limiting factors characterised the Early Holocene. Furthermore, our model reveals geographic differences in the limiting factors between Fennoscandia, Southern, Central, and Eastern Europe. The spatiotemporal variation in limiting factors suggests that hunter-gatherers needed to overcome very different adaptive challenges in different parts of Europe across this period of climatic and environmental change.

## Results and discussion

### Models of hunter-gatherers' population density

The relation between the environmental factors explored here (Table 1) and population density assessed using qGAMs (Fig. 1) was negative for temperature seasonality (TS); positive for effective temperature (ET), net primary productivity (NPP), and temperature of the coldest month (MCM); unimodal for the temperature of the warmest month (MWM); and asymptotic for potential evapotranspiration (PET). Total annual precipitation (TAP) and monthly limits showed an overall flat trend. In all cases, our qGAMs performed significantly better than a model without predictors (hereafter termed the mean model), as determined by the significant deviance explained (Table 1).

In our 50th percentile qGAMs, most of the environmental variables produced models that explain, on average, between 10% and 49% of the population density variation among ethnographic foraging societies (Table 1). The five environmental variables with the highest model performance compared to a mean model (deviance explained in Table 1; see Methods) were MCM, TS, ET, NPP and MWM. MCM represents the effect of yearly limiting conditions (= winter mortality) on ecological performance and hence demographic trends[34,35]. The other two temperature variables (ET, TS) relate to spatiotemporal energy variation[34]. Lastly, net primary productivity (NPP) reflects environmental productivity in the form of the availability of plant resources, where high values lead to larger population densities, as already suggested by a plethora of earlier studies[36–38]. These variables display high correlations (Pearson correlations range between 0.83 and 0.96), supporting the notion that they broadly reflect energy availability and variability effects on forager demography. Other variables related to environmental productivity (PET) have lower yet somewhat similar predictive accuracy (Table 1). We use the correlation between predictors to underpin our grouping of individual variables within suites of possible explanatory mechanisms (as listed in Table 1). This correlation between variables is not in itself a problem in our analytical paradigm as we are building univariate models. Furthermore, as our analyses aim to define the possible mechanism acting as a limiting constraint, considering the absolute effect of individual variables at a time allows us to capture their effects better.

Besides the well-known limitations of using foragers of the recent past for reconstructing prehistoric social and demographic conditions[13], the issues of model truncation and non-analogy of climatic conditions present themselves as major potential caveats. Climatic non-analogy here refers to the problem of projecting models beyond the domain for which they have been calibrated[39–41]. Model truncation refers to the incomplete characterisation of hunter-gatherer populations' total climate space[42–44] and has been a long-noted limitation of ethnographic analogies for prehistoric foragers[45]. Yet, it has already been shown that the dataset assembled by Binford is not critically biased in terms of forager niche space[26]. However, we see that MWM, under current conditions, does not fully represent the environmental states of the evaluated period's early (Pleistocene) time

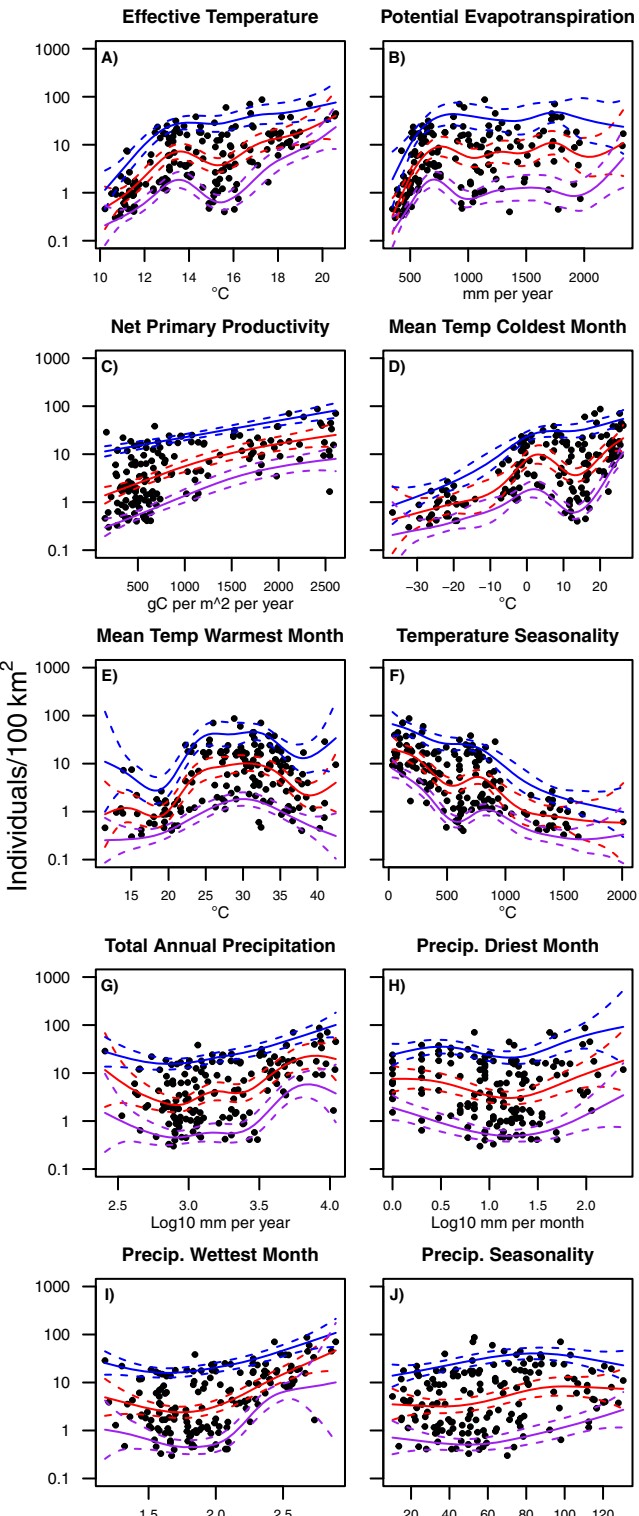

**Fig. 1 | Quantile Generalised Additive Models (qGAM) describe the relationship between environmental factors and population density.** Quantile Generalised Additive Models (qGAM) describing the relation between environmental factors and population density for **A** Effective temperature, **B** Potential evapotranspiration, **C** Net primary productivity, **D** Mean temperature of the coldest month, **E** Mean temperature warmest month, **F** Temperature seasonality, **G**) Total annual precipitation, **H** Precipitation driest month, **I** Precipitation wettest month and **J** Precipitation seasonality. Lines on each panel show the 10th percentiles (purple lines), 50th percentiles (red lines), and 90th percentiles (blue lines). Solid lines show the mean predicted values, and dashed bands indicate the 95% confidence intervals of each model prediction.

points. Therefore, MWM was not considered in our population density estimation and limiting factors. For the other variables, we do not see either truncation or severe non-analogy: the climate space observed at different moments during the 21-8kyBP period shows broad overlaps with the climate space used to develop our qGAMs (Fig. 2). Therefore, our models are not unduly extrapolating into environmental regions where there is no clear indication of how population density changes as a function of evaluated climatic variables. By the same token, it is necessary to highlight that the distributions of some palaeoclimatic conditions—including all those with the highest predictive values in our models—are skewed towards the lower end of contemporary values. This skewness is especially pronounced for the Pleistocene and variables such as maximum temperatures (Fig. 2), affecting our inferential power on changes in population densities at these extremes. Therefore, our hindcast population densities are gross over-estimations, especially for the Pleistocene, where temperature-related variables dominate as limiting factors.

## Human populations densities across the Pleistocene-Holocene transition

Based on our 50th percentile qGAM models, Europe's estimated climate-limited human population size was smallest at 21kyBP (~117,500 individuals) and largest at 8kyBP (~213,900 individuals). Based on the 90th and 10th percentile models, these climate-limited human population sizes could have been as high as ~625,000 individuals and as low as ~28,300 individuals at 21kyBP. At 8kyBP, a climatically limited population size could have been as high as ~1,111,000 individuals and as low as ~52,000 individuals. These estimates represent the overall continental-scale human population size to be expected if climatic conditions were the only factor affecting the number of individuals and under conditions where all available space was, in fact, occupied.

Also, based on our 50th percentile qGAM model and using a viable population density threshold of 0.2 individuals/100 km², we show that at the warmest point of Greenland Interstadial 1 (~14.7kyBP; GI1), Europe's human population size estimated by our model was ~155,000 individuals; a number that decreased to ~143,000 individuals at the coldest point of Greenland Stadial 1 (~11.7kyBP; GS1). The overall occupied area (number of inhabited cells), based on our 50th percentile qGAM model, was 62% of the ice-free region at the end of the GS2 (~21kyBP), increasing to ~74% during GI1 and reaching its nadir (~93%) by 8kyBP. Forager land-use was evidently extensive, however, including many largely empty spaces[46]. Still, at continental and centennial-to-millennial scales, overall population growth is suggested. As indicated by a strong correlation (Pearson's rho = 0.7 $p$ = 0.007)—and despite the differences in the temporal aggregation and types of data utilised—there is a robust alignment in trends between our estimates of population size and independently derived archaeological occupation proxies (red lines; Fig. 3).

During the evaluated period, the mean population density in the inhabited area varied between 1.6 and 1.8 persons per 100km² (GS2 = 1.63 p/100 km²; GI1 = 1.70 p/100 km²; GS1 = 1.61p/100 km²; EHol = 1.78 p/100 km²; Fig. 3). Mean temporal patterns were similar to those predicted based on 90th and 10th percentile qGAM models (Supplementary Fig. 1). Although the temporal patterns in average population density derived from our limiting-factor analysis are similar to those of core area estimates by ref. 24 (blue areas, Fig. 3), these do not match numerically due to our focus on 50th percentile population densities. Our population density estimates are consistent with those suggested by ref. 12, and more recently by ref. 11.

The estimated pattern of human population density (Fig. 3) indicates a population expansion starting almost 3ky after the ice sheet began to recede from its maximum extent at 21kyBP. Evaluating the spatially explicit predictions of our model (Fig. 4 and Supplementary

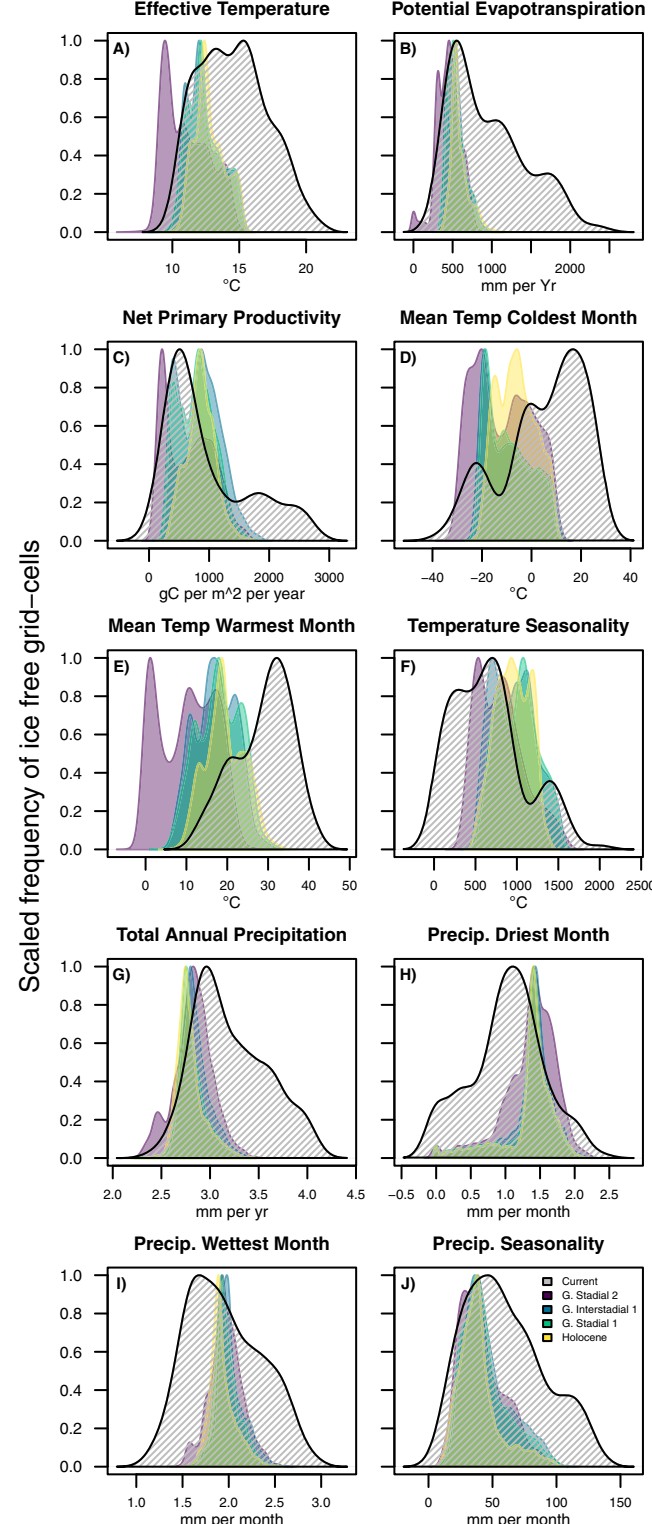

**Fig. 2 | Distribution of climatic conditions today and selected periods during the Pleistocene-Holocene transition.** Convergence between current climatic conditions (hashed density plots) and paleoclimatic conditions at four different periods (coloured density plots) for **A** Effective temperature, **B** Potential evapotranspiration, **C** Net primary productivity, **D** Mean temperature of the coldest month, **E** Mean temperature warmest month, **F** Temperature seasonality, **G** Total annual precipitation, **H** Precipitation driest month, **I** Precipitation wettest month and **J** Precipitation seasonality. Paleoclimatic periods are Greenland Stadial 2, Greenland Interstadial 1, Greenland Stadial 1 and Early Holocene. Panels show the density plots for each evaluated variable (Variables are explained in Table 1).

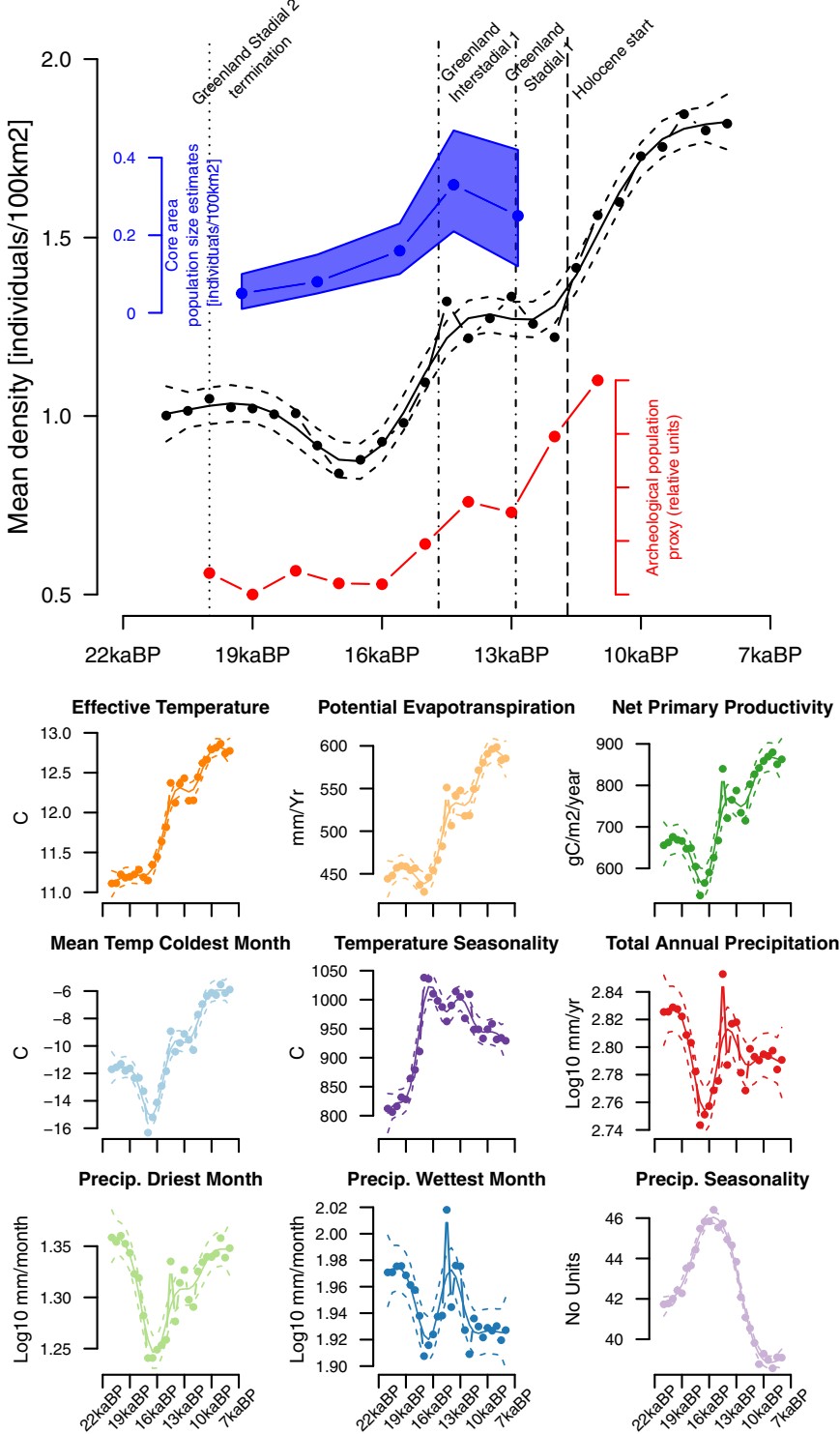

**Fig. 3 | Estimates of climatically possible mean population density (top) and trends in key environmental variables (bottom).** Estimated average population density for all of Europe is based on the average of grid-based population densities (minimum predicted density for a cell) in ice-free regions (black line−top panel) based on the 50th percentile generalised additive models (qGAM). Here solid black lines show the mean predicted values and black dashed bands indicate each model prediction's 95% confidence intervals. Estimates for 90th and 10th quantiles are presented in Supplementary Fig. 1. These estimates are compared to archaeological population proxy based on the number of calibrated radiocarbon dates for Europe between 21 and 11kyBP extracted from ref. 12. Summaries of the Radiocarbon Palaeolithic Europe Database v28[105] (red−top panel), and core area calculations (cf.[24] population density mean [solid blue line] and upper/lower estimates [blue area limits] based on the Cologne Protocol (blue−top panel). The bottom panels show the changes in continental averages of the environmental variables used to determine our population density estimate (minimum across all variables). For each environmental predictor, solid lines show the continental mean, and dashed bands indicate the 95% confidence intervals.

# Individuals per 100 km²

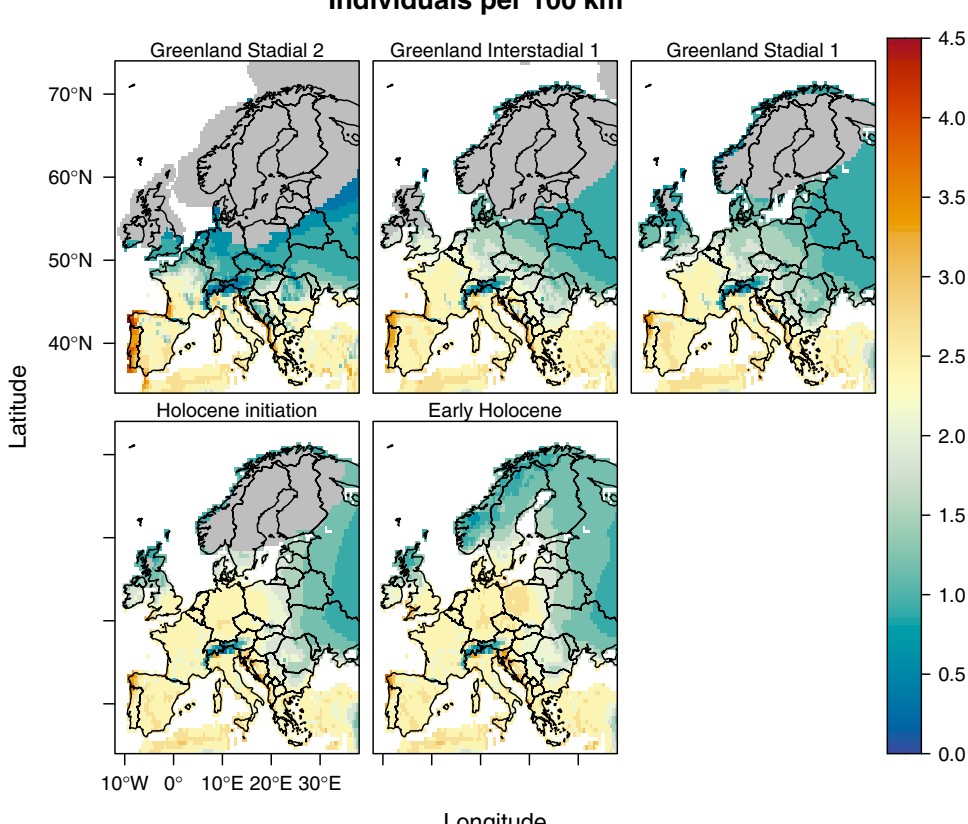

**Fig. 4 | Estimated human population density (persons/100 km²) across Europe for selected times during the 21ky to 8kyBP period.** Predictions are based on 50th percentile generalised additive models (qGAM), and any human presence is coloured. Areas in grey represent the glacier extent as derived by ICE-6G-C[104]. Panels from top to bottom and left to right are Greenland Stadial-2, Greenland Interstadial-1, Greenland Stadial-1, Holocene initiation and Early Holocene.

Figs. 2, 3), we find that at the end of the GS2, hunter-gatherer societies in Europe extended—at appreciable potential densities—as far north as central France, southern Germany and southern parts of modern-day Ukraine with steep negative gradients northwards. This distribution pattern is consistent with the archaeological evidence for the recolonisation of Europe[47–50]. Our models also suggest that by the end of the GS2, a relatively large proportion of the European continent may have been at least sporadically visited (~62%; Fig. 4A, B), with the Mediterranean region up to the northern Alpine foreland showing population densities up to five individuals/100 km². This restricted occurrence pattern is also supported by the archaeological record[46]. Our model also indicates a persistent southwest-northeast gradient of decreasing population densities in this southern region, with the most populated areas occurring in the Iberian Peninsula specifically and the Mediterranean region more broadly (Fig. 4A, B). Recent archaeological analyses suggest early exploratory dispersals northwards between 19–18kyBP[51], which may correspond to the slight increase in modelled population densities. Subsequently, a more sustained recolonisation of the continent gathered pace from ~17kyBP (Fig. 4A), reaching up to Scandinavia by the onset of GS1 (~12.8kyBP, Fig. 4C). Earlier archaeological[52,53] and modelling studies[23] have already suggested that this colonisation was rapid but also that it proceeded in several steps where both climate and landforms served as barriers to expansion[23]. Our results contribute to this discussion by highlighting that different climate variables limited human dispersal for a given location and that these limits changed over time.

## Limiting factors of population density estimates

Using our limiting factor approach, we improve our understanding of demographic mechanisms in Late Pleistocene and Early Holocene European hunter-gatherer societies by highlighting the spatiotemporal changes in the main factor restricting population density (Figs. 5, 6 and Supplementary Fig. 4). Our modelled population density estimates can be linked to regional or local narratives or empirical tests of changes in occurrences and population sizes (e.g. refs. 26, 54). The changes in limiting factors suggested in our models can be divided into three periods. The first period spans from the termination of GS2 to the onset of interstadial warming at around 15kyBP. During this period, energy availability measured as ET was the main factor limiting population density across most of Europe (~30% of cells; Figs. 5 and 6A). Likewise, TAP was also a strong limiting factor (~30% of cells; Figs. 5, 6A). Furthermore, limitations imposed by winter temperatures could also be considered as likely limiting factors based on estimates of average conditions on a continental scale (Fig. 6B). The range of experienced temperature conditions, represented by ET, can thus be seen as the major limiting factor shaping human population density in Europe between GS2 and the initiation of warming associated with GI1 (Figs. 5, 6A). With temperature-related variables as the overwhelming limiting factor during this period, the emergence of sophisticated sewing techniques[55] and pyrotechnology[56] likely facilitated the persistence and even moderate expansion of populations at this time.

The second period covers the rapid warming (GI1) as well as cooling (GS1) events between 14.7kyBP to 11.7kyBP. During this period, the overall importance of ET steadily decreased, with both TS (~32% of cells) and TAP becoming the main factors limiting population density (Figs. 5 and 6A). Limitations imposed by winter temperatures remained in force in some parts of central Europe (Fig. 5). The decrease of ET as a limiting factor indicates that during this period of rapid change, it was not temperature but its variability (measured as TS) and energy availability (due to the link between PET/TAP and

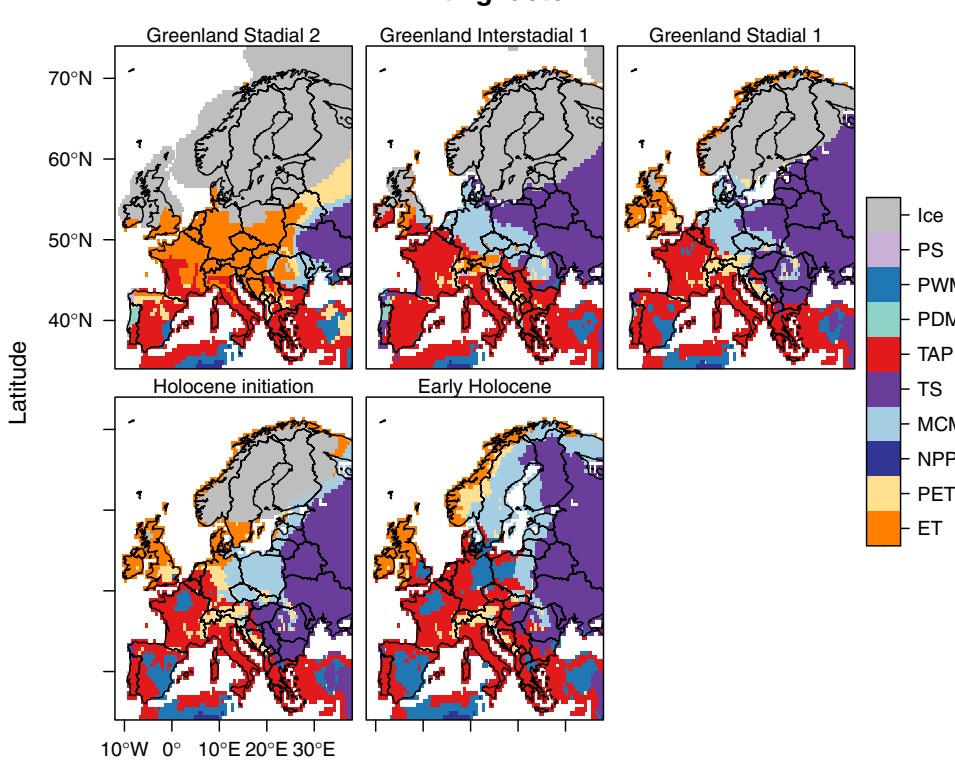

**Fig. 5 | Factors limiting population density across Europe for selected times during the 21ky to 8kyBP period.** Predictions based on 50th percentile generalised additive models (qGAM). Areas in grey represent the glacier extent as derived by ICE-6G-C[104]. Panels from top to bottom and left to right are Greenland Stadial-2, Greenland Interstadial-1, Greenland Stadial-1, Holocene initiation and Early Holocene.

productivity; refs. [57], [58]) that determined human population density in Europe. Our models suggest that overall population densities increased (Fig. 3), although a temporary reduction associated with GS1 cooling also stands clear.

The last period encompasses the early part of the Holocene from its onset at 11.7ky to 8kyBP. Here, TS and TAP were the main limiting factors (~60% of cells; Fig. 6A), while the effect of ET became marginal (Fig. 6A). These patterns indicate a complete shift from experienced temperature conditions to annual variability and available resources as the main limiting factors of European forager population densities during the Holocene. Such a shift is interesting as the Early Holocene also witnessed a significant reorganisation of forager socio-ecological systems towards more varied use of resources and more pronounced territoriality focused on spatially circumscribed and regionally available resources. It also saw a widespread shift from immediate-return to delayed-return economies increasingly characterised by a focus on food storage through smoking, roasting and fermentation that necessitated considerable investments in time and resources[59–61]. Furthermore, this shift also aligns with the idea that decreasing territory sizes and more marked boundary formation directly relate to the spatiotemporal dynamics of resource availability[62].

The regional disaggregation of patterns in limiting factors shows strong differences between Fennoscandia, Southern, Central, and Eastern Europe (Figs. 4 and 5). These patterns persist over time, with regional shifts linked to temperature change as a key feature. In Fennoscandia and the British Isles, ET was the main limiting factor for most of the Late Pleistocene. This pattern changed after the onset of the Holocene when TS and TAP became the dominant limiting factors. In Eastern and Western Europe, ET was the main limiting factor at the end of the GS2 but was replaced by TAP in the west and TS at the onset of the GI1. During GS1 and the Early Holocene, the main limiting factors

were TAP and TS. In southern Europe, especially in the Mediterranean, TAP was the main limiting factor from GS2 onwards, supporting the idea that humans were closely tied to water resources in mid-latitudes[63].

Our analyses show that the main limiting factors constraining forager population densities across the Last Glacial-Interglacial Transition in Europe changed markedly over space (Fig. 5) and time (Fig. 6A). With these detailed dynamics in hand, we can now return to the archaeological record in search of material culture proxies that may have allowed these past communities to overcome these limiting factors[64–66]. These may have related to water availability (= containers) in the Mediterranean and are predicted to relate to temperature (= clothing or pyrotechnology) in higher latitudes. Certainly, for the latter two technologies, recent analyses suggest that the diversity and complexity of sewing technologies[55], shelters[67] and fireplace construction[56] and use[68,69] dates to the periods in which temperature-related variables acted as limiting factors. Conversely, where such technologies are absent in the archaeological record, we can also begin to think about population vulnerability to climatic factors at regional levels. Especially in higher latitudes, population fluctuations may have been pronounced at the sub-centennial scale, to the point of local population extirpations[9,70].

Finally, the marked shift in limiting factors at the onset of the Holocene may be reflected in the observed shift away from energy and heat conservation technologies towards resource access and harvesting behaviours and technologies via processing, and that the increased spatial heterogeneity of limiting factors also engendered stronger cultural differentiation at a regional scale. The spatiotemporal dynamics of resource availability directly impact land-use, mobility, territoriality and the formation of information networks in foragers[62,71]. In line with these expectations, regional cultural signatures became

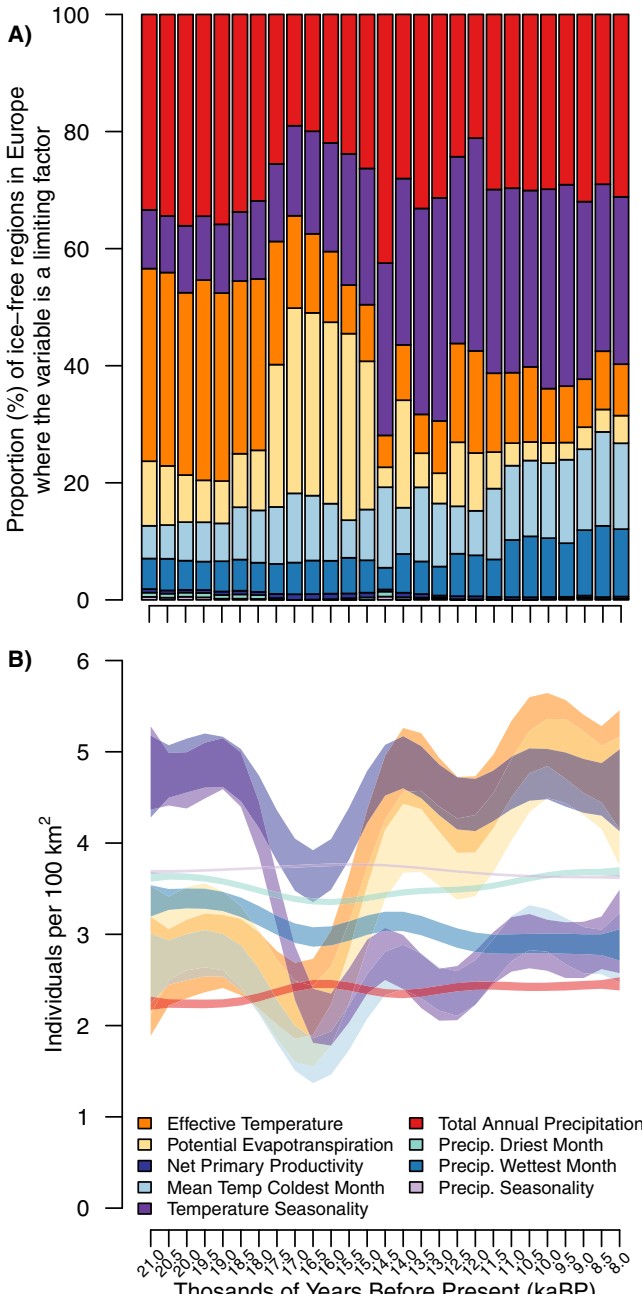

**Fig. 6 | Summary of European-wide limiting factors to population density during the Pleistocene-Holocene transition.** Proportion of the ice-free area of Europe where each variable was estimated to be the factor limiting population density (**A**) and estimated population size based on the mean environmental conditions for each century (**B**).

vary. Our approach offers a way to infer the hierarchy of limiting factors and hence provides a spatiotemporal hypothesis for major selection pressures acting on forager populations in the past.

Independent palaeodemographic estimates broadly support our models, but many questions remain. Climate models, for instance, only indirectly capture the interaction of human population dynamics with changes in biodiversity and ecosystem compositions. In addition, the match between modelled population densities and the field-validated presence of Late Pleistocene/Early Holocene populations is not equally robust everywhere. These deviations may stimulate targeted field-testing that assesses whether and why population densities periodically fell short of or exceeded modelled values. In conjunction with legacy data derived from archives and the literature, such fieldwork can also shed light on the specific strategies these past foragers employed to mitigate the risks posed by specific limiting factors.

Small-scale societies have various adaptive options at their disposal (see ref. 72), most of which can be captured through archaeological proxies[73–75]. Our limiting factor model provides an explicit spatiotemporal hypothesis as to which risk mitigation measures could have been in use at any given time and place. By comparing our climate-driven estimates of population density to those estimated using the archaeological record we can identify potential mismatches suggestive of declines or increases not predicted by our models. By then identifying if shelters, fire or projectile technology, among other technologies, provided a given group with the capacity to buffer and perhaps even overcome climatically determined demographic limitations, we can better assess the adaptive role of specific cultural adaptations. The successful testing of such hypotheses would then shed light on these populations' resilience and adaptation—or lack of it—during this climatically and environmentally tumultuous time. Finally, the marked shifts in dominant limiting factors identified in our models map onto results of Late Pleistocene/Early Holocene Earth System tipping points recently discussed by Brovkin and colleagues[76]. Analogous to anthropogenic warming in the present, these periods of rapid and substantive climatic change are likely to have created challenges for contemporaneous forager populations. In an effort to align archaeological perspectives on climate change with the quandaries of our time (cf. ref. 77), future research would be well-advised to focus on such periods of major systemic transitions.

## Methods
### Models of hunter-gatherers' population density
We use ethnographic data on terrestrially adapted, mobile hunter-gatherers and their climatic space[26] to construct a series of statistical models that predict hunter-gatherer population density based on one of ten climatic predictors (see Table 1 for rezoning and source). While there are important caveats[13], this approach builds on multiple ethnographic studies showing a link between climate on the one hand and hunter-gatherer diet, mobility, and demography on the other[14,71,78–82]. This statistical connection is the basis of recent studies focused on building complex multivariate models of population dynamics[11,12,29]. A benefit of our statistical approach is that it overcomes some significant limitations, such as lack of quantitative population size data based on the archaeological record itself or genetic data, each associated with its own limitations (as reviewed in refs. 2, 12,). Also, this univariate approach allows us to evaluate the absolute effects of multiple correlated environmental variables in the same study. Due to the nature of this study, it was not subject to ethics approval.

We omitted four observation classes in the original ethnographic dataset in defining the association between hunter-gatherer population density and climatic predictors. First, we removed observations associated with food producers. Second, sedentary populations or those that reside at a single location for >1 year. Third, populations using aquatic resources (>30% of their dietary protein comes from aquatic environments, as defined in refs. 83, 84,). Forth, we excluded

more pronounced in the Holocene, and borders between different cultural zones were eventually more strongly articulated. These patterns could be seen as a response to the fundamental shift in limiting factors we have identified in our models.

### Placing limiting factor analyses in an archaeological context
Seeking correlations between environmental variables and past human population densities is not a new endeavour. Following recent calls for more theoretically-informed rather than mere statistical explorations of this relationship[13], we highlight that while the environment can be said to strongly constrain forager lifeways, precisely which aspects of the environment do so at any one place and time must be expected to

all observations related to horse-riding populations. The filters employed here correspond to those used by ref. 12 to maximise the match between ethnographic data and the current knowledge of the highly mobile and overwhelmingly terrestrially oriented lifestyles of Late Pleistocene/Early Holocene hunter-gatherers in Europe. The implemented filters are less restrictive than those used by other studies that have sought to reconstruct forager population dynamics during this time[85] and thus allow for a relatively large degree of behavioural variation. This is important given that increasing evidence of marine and lacustrine resource use is emerging for at least certain times and regions in Late Pleistocene Europe[86–88], and that a marked diversification characterises the resource base of Early Holocene foragers. Finally, these filters remove any population using external supplements to their hunter-gatherer lifestyle, resulting in a database including information on 159 populations.

We used climate data on historical averages (1970–2000) for 19 climate variables (Table 1) to build our ethnography-based population density models. These were obtained from the Worldclim version 2.1[89] at a 10-ArcMin resolution. Importantly, we used Worldclim data instead of climatic variables directly available from Binford's dataset to ensure comparability between climatic variables not in the database (i.e. net primary productivity, yearly limits). Equally importantly, this approach prevents any estimation biases due to differences between the data used to define climate-density relations and paleoclimatic surfaces used to estimate population density changes and limiting factors over time.

Initially, we model how population densities of hunter-gatherer communities change along current environmental gradients using quantile generalised additive models (qGAMs). Modelling such dynamics using qGAMs offers a transparent way to determine the nonlinear changes in different percentiles of a response variable (= population densities) to one or multiple environmental variables and to understand the response-predictor dynamics outside of the mean of the data, making it useful in understanding outcomes that are non-normally distributed and that have nonlinear relationships with predictor variables[30,31,90]. This approach is commonly used in the ecological literature to determine the likelihood of occurrence or abundance of a given species under a particular environmental regime[91–95] but has never before been applied to human palaeodemography.

In contrast to previous studies evaluating past human population density changes, we do not consider the synergies between multiple climatic variables when describing the relation between population densities and climate. We reasoned that using multiple predictors simultaneously in one model results in the evaluation of relative effects, and that these are contingent on the variables used in the model. Furthermore, these relative effects do not capture the limiting effects exerted by a given variable, but the partial contributions of each evaluated variable to the deviation from the regional mean population density. We therefore instead focus on the individual effects of evaluated variables on different percentiles of population densities (90th, 50th and 10th percentiles) to (i) identify the most pronounced limiting factor determining palaeodemographic patterns and (ii) assess how the climatically possible maximum, average and minimum population densities changed in space and time. The shapes of the relation between population densities as a function of environmental variables were consistent for different percentiles (Fig. 1).

The population density estimates derived from the ethnographic data followed a log-normal distribution, so these were log transformed for subsequent analyses, and a gaussian response distribution was used in our qGAMs models. Annual and yearly limit precipitation variables were similarly transformed. The ability of each of the evaluated variables to predict hunter-gatherer population densities was determined using the mean deviance explained (1 − (residual deviance/ null deviance)), which in the context of this study is identical to a GLM

unadjusted $R^2$ [96,97]. These were calculated both for the whole dataset and using a 1000-fold cross-validation approach. We used an h-block approach[32] to select our training and testing datasets to avoid any possible spatial autocorrelation in the datasets, as Binford's ethnographic database is known to consist of populations some of which are ecologically, demographically, and culturally related. Our h-block design uses the range of the population density variogram to define the minimum distance two observations can be apart (4500 km) to be included in our training dataset. All models and prediction accuracy estimates were implemented in R (version 3.6[98]) using the mgcv (version 1.8.24[99]) and qgam (version 1.3.2[100]) packages. Code to replicate these analyses is available from[101].

### Estimating human populations density, size and limiting factors across the Pleistocene-Holocene transition

The monthly average temperature and annual precipitation values for Europe for the 21ky to 8kyBP period come from the CCSM3 SynTrace paleoclimate simulations[102]. These were bias-corrected and downscaled to 0.5° × 0.5° following the methods described by ref. 103. The paleoclimatic simulation data used here were originally generated to evaluate changes in European and North American fossil pollen data and vegetation novelty since the Last Glacial Maximum[33]. Source climate surfaces were aggregated to 500-years intervals from the original decadal averages of monthly values. We only considered locations not covered by ice, using a mask from the ICE-6G gridded data product[104]. As CCSM3 SynTrace paleoclimate simulations do not contain estimates of uncertainty, we cannot assess how variability in paleoclimatic model outcomes would propagate to our estimates of population density and limiting factors.

To generate population density estimates for each evaluated percentile within each variable/500-year interval, we used the average of the predictions resulting from running the 1000 h-blocked qGAM models generated for each variable under the conditions of each 500-year interval. To mitigate any truncation or non-analogy artefacts, only those qGAM models where all historical means were contained under current conditions were projected into past climatic conditions. Population density for each cell within each 500-year interval point in time is then determined as the minimum predicted density for that cell. To calculate the human population size in Europe during every century, we multiplied the predicted population density in each cell where the predicted population density was above 0.2 individuals per 100 km² (the lowest densities recorded in the ethnographic dataset) by the land area of the corresponding cell to arrive at per cell population size. We then summed these values to arrive at the total population size for each century. As our objective was to establish the climatic variable that imposed the strongest constraints on hunter-gatherer population density at any one time, we determined for each of the three evaluated percentiles the variable estimating the lowest population density for a given cell at each evaluated time-period to be the limiting factor (the scarcest resource that would then limit population size cf. ref. 17). For each evaluated time-period, we summarised the proportion of the available land area (i.e. land area not covered by ice) where each of the assessed variables was determined to be the limiting factor.

Uncertainties in population density, size, occupied area, and limiting factor estimates were determined using a cross-validation approach, where model fitting was iterated 1000 times using a random sample (70%) of the ethnographic and climate data at each time step. Each model was used to hindcast population densities, estimate the percentage of inhabited land area and human population size, and define the relevant limiting factor. Uncertainty in continental-scale estimates of population densities, occupied area and population size was determined using 95% confidence intervals. The variable selected as the limiting factor in most cross-validation folds was selected as the limiting factor.

## Validation of population density estimates

To assess the validity of our population density estimations, we use the International Union for Quaternary Science (INQUA) Radiocarbon Palaeolithic Europe Database v28[105]. Changes in the density of records are a useful continental-scale proxy measurement of prehistoric population size changes are ommonly used to describe prehistoric human population dynamics trends[106–111]. We extracted $^{14}C$ dates from the INQUA Radiocarbon Palaeolithic Europe Database, aggregating these to the closest 1000 years in order to determine the match between our qGAM-derived populations' density estimates and those derived from the frequencies of radiocarbon dates between 21kyBP and 10kyBP, closely following ref. 12. This approach allowed validating our hindcasted estimates of absolute prehistoric population density since our model is not archaeologically informed, avoiding any possible circularity between model development and validation.

We also used site-based estimates of population density as derived using the Cologne Protocol by ref. 24. We focus on estimates of extended interconnected socio-economic areas (Core Areas) for five unequal time bands between 25kyBP and 11.7KyBP. Although ultimately also based on Binford[26], these estimates present independently derived estimates of population density for Late Pleistocene populations in Europe.

All maps were generated using R (version 3.6; ref. 98) and the packages raster[112] (version 3.4-13) and maptools[113] (version 1.1-2). Code to replicate these is available from ref. 101.

## Reporting summary

Further information on research design is available in the Nature Research Reporting Summary linked to this article.

## Data availability

All the datasets used in this study are publicly available. The 'Binford' ethnographic database[26] is available from the Database of Places, Language, Culture and Environment (D-PLACE; https://d-place.org/contributions/Binford). Current and Late Quaternary environmental datasets are publicly available from the associated references. International Union for Quaternary Science (INQUA) Radiocarbon Palaeolithic Europe Database v28 is available from https://pandoradata.earth/dataset/radiocarbon-palaeolithic-europe-database-v28. Contemporary climate databases are available from WorldClim V.2.1. project (https://www.worldclim.org/data/worldclim21.html) and Late Pleistocene climate sources are available at https://doi.org/10.6084/m9.figshare.c.4673120.v2. Source data are provided with this paper.

## Code availability

Code and subset of the data used in this study is available at https://doi.org/10.5281/zenodo.6962693 [101].

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

## Acknowledgements

A.O. was supported by the AUFF Starting Grant (AUFF-F-2018-7-8). F.R.'s contribution is part of CLIOARCH, an ERC Consolidator Grant project that has received funding from the European Research Council (ERC) under the European Union's Horizon 2020 research and innovation programme (grant agreement No. 817564).

## Author contributions

A.O.: Conceptualisation; Methodology; Formal analysis; Resources; writing—original draft, writing—review and editing; Visualisation. F.R.: Conceptualisation; Methodology; writing—original draft, writing—review and editing.

## Competing interests

The authors declare no competing interests.

## Additional information

**Supplementary information** The online version contains

supplementary material available at

Alejandro Ordonez.

**Peer review information** *Nature Communications* thanks Ron Pinhasi,
Miikka Tallavaara and Dan Zhu for their contribution to the peer review of
this work. Peer reviewer reports are available.

