## [Peer Review File · Nature Communications]

Changes in limiting factors for forager population dynamics in Europe across the Last Glacial-Interglacial TransitionReviewers' Comments:

Reviewer #1:

Remarks to the Author:

This study aims to identify the limiting climate factors of hunter-gatherer population density in Europe from the LGM to early Holocene, which is a tempting research question that, if answered convincingly, could provide clues in resilience and adaptation strategies of ancient human societies. The authors built statistical models (quantile Generalised Additive Models) of population density versus each of the 18 climate variables based on contemporary hunter-gatherer dataset, and hindcasted population density using paleoclimate outputs from a climate model. This workflow and statistical techniques are not new (e.g. Tallavaara et al. 2015 pnas), but selecting the factor that predicts the lowest population density across space and time, based on the concept of limiting factor, provides a fresh angle of view. However, I have some major concerns about the robustness of the analysis and significance of current results, as outlined below. Based on these, I cannot recommend its publication, at least not in its current form.

Robustness of results:

Mean temperature of the Warmest Month (MWM) is identified a major limiting factor during all critical periods (Fig. 4), but MWM is the variable that has the most severe non-analogy problem comparing present-day climate space and past climates (Fig. 2 and Supplementary material S2). Although mentioned at Lines 126-128, considering the strong relevance to the main findings, the risk of an unreliable extrapolation out of the range of data used to fit the statistical model is higher than acknowledged here.

In addition, it is not clear to me why the authors chose 90th percentile of population density to do the hindcast. First, the results, in principle, would not be comparable to previous estimates in literature. Second, does the resulted limiting factors change dependent on the choice of the percentile? Although the general shape of the population density versus climate relationship looks similar across different percentiles for each individual climate factor (Lines 311-312), it is the relative magnitudes between all predicted densities by these factors that ultimately selects the limiting factor. Thus, it is not straightforward whether your results are sensitive to the choice of percentiles.

Regarding the comparison between hindcasted population density and the archeological population proxy (Fig. 3), I would not say they are "in line with" each other (Lines 139-140). The black curve in Fig. 3 starts to increase already since 18 ka, which is relatively flat in the red curve; the red curve increases significantly during GI1 and GS1, whereas it is stale in the black curve.

By the way, at Lines 101-103, why do you separate higher and lower predictive accuracy by a threshold of "explained deviances < 0.79"? According to Table 1, these predictors are all so close.

About mechanism insights:

From the results it is hard to infer mechanisms regarding how the identified limiting factor has constrained population density. This is limited by the fact that only temperature and precipitation and their variants were used as predictors, without direct information about productivity; whereas climate impacts population density via indirect effects on ecosystem attributes like NPP (e.g. Freeman et al. 2020, doi:10.1016/j.jas.2020.105168). Throughout the text the authors have tried to relate some of the factors to environmental productivity, but it was highly speculative. Let me take Lines 185-192 as an example. During 14.7ka to 11.7ka, the importance of ET decreases while importance of MWM and temperature seasonality increases. But all three variables are linked to NPP (and possibly other aspects of the ecosystem). From these changes one still cannot judge what process is taking effect in the end.

Given this, why not use NPP as a predictor in the first place? Data availability for the hindcast should not be a problem since simulated NPP for the past 21,000 years are publicly accessible from some climate models already.

Significance of the results in archaeological perspective:

I commend the authors' effort to put the (more of ecology-oriented) results into archaeological

context, but currently it is still limited in qualitative descriptions scattered in the text. If the authors could achieve a more systematic compilation of archaeological records regarding how these societies have tackled with the limiting climate factors and find a consistency with your hindcasted results in space and time, it would add much merit to this study, with broader significance and impacts.

Aside from the above concerns, the organization of Results and discussion needs to be improved. Adding sub-headings would help. Besides, descriptions of the results in the text should be more careful. Currently they are sometimes inconsistent with the table or figures. For example, at Lines 104-110, it says seasonal temperature variables are among the lowest explained deviance, which is not the case as listed in Table 1.

A minor point is that the uncertainties/biases in the paleoclimate outputs of the CCSM3 climate model should be discussed.

Code availability: though not mandatory, it is strongly encouraged to make the code readily available so as to enable reproduction of the results.

Table 1: A conceptual confusion: MCM is not "extreme events". Same for MWM, PDM, and PWM. Extreme events are events that occur with low frequency, not the regular seasonal maxima or minima.

There are a few careless errors in the manuscript, for example:

Line 266: "16 climatic predictors": there are 18 climate variables in Table 1.

Line 289: it says "We use a subsample of 159 hunter-gatherers populations..." in the Reporting summary, while here it says "127 populations".

Table 1: Acronym of "Precipitation of the Wettest Month" should be PWM, not PDM.

Figure caption of Fig. 3: "Minimum temperature of the Coldest Month" should be "Mean...", and "Maximum temperature of the Warmest Month" should be "Mean..."

Figure caption of Supplementary material S1: it is not "the six most important environmental factors". Please check carefully throughout the paper.

Reviewer #2:

Remarks to the Author:

This is truly thought-provoking, highly interesting and novel contribution to the hunter-gatherer ecology. I very much like the approach of applying the analysis of ecological limiting factors, for the first time, to prehistoric hunter-gatherers. Below I have highlighted few issues that you could consider to revise to further improve the paper.

Best regards,
Miikka Tallavaara

1.

Given that many of the climate predictors are highly correlated (as you also write in the manuscript), they will convey partly the same information. This can potentially make it difficult to differentiate the importance between different climatic variables as limiting factors. I would suggest that you add some more justification for using large number correlated variables in the analysis. Or, alternatively, consider reducing the dimensionality of the data.

2.

Partly related to the item 1, you could provide justification for using univariate instead of multivariate approach. The effect of a predictor variable can change (sometimes dramatically) when controlled for the effects of other variables by adding them to the model. Therefore, you should explain why you rely on univariate approach, or, alternatively, try to add the best predictor candidates in the same model

and see how the results would change.

3.

Binford's data is notorious for spatial auto-correlation, especially because in particular areas, he has basically split closely living ecologically, demographically and culturally similar groups into smaller units even though one might consider many of those belonging to the same ethnic group. This can lead to inflated performance metrics in traditional cross-validation schemes. The idea of cross-validation is to test the model with data that the model has not seen before, so in the presence of spatial auto-correlation, test data can be "too" similar to training data. Therefore, your performance metrics are quite likely "too good" and I suggest that you could use some kind of spatial block cross-validation scheme, such as h-block cross validation. See, e.g.

Salonen, J.S., et al, 2016. Calibrating aquatic microfossil proxies with regression-tree ensembles: Cross-validation with modern chironomid and diatom data. *The Holocene* 26, 1040–1048.

<https://doi.org/10.1177/0959683616632881>

<https://quantpalaeo.wordpress.com/2013/12/15/h-block-cross-validation-of-transfer-functions/>

https://cran.r-project.org/web/packages/blockCV/vignettes/BlockCV_for_SDM.html

4.

Related to the above issue, on page 3 you write that no single environmental variable explained more than 81% of the population density variation among ethnographic foraging societies. It might be because of my ignorance of quantile regression, but I'm not sure if you can really say that quantile regression model can explain some percentage of the variation in a response variable. So clarify this and explain what the explained deviance is measuring in your quantile regressions, is it the goodness of fit of the 90th quantile or what?

I'm also not sure if one can directly compare your performance metrics (explained deviance) to e.g. our metrics (R^2) (Tallavaara et al. 2015). Besides, our R^2 e.g. for multivariate GAM is clearly smaller (0.6), not marginally better, than any your values. After a lot of exploration with Binford's data, I also think that it is really difficult to push the R^2 of (multivariate) population density models well above 0.7 unless you really overfit the model.

5.

On page 5, you provide the modelled population size estimates for Europe, which seems to be pretty high. The LGM estimate is twice as large as our previous estimate (which has been argued to be way too large by some) despite we having larger geographical area. However, am I right that your estimates are actually maximum estimates based on the modelled 90th quantile? Whatever the case, this needs to be stated clearly in the text and in the relevant figure captions.

6.

I might have missed it somehow, but which of the many univariate models you are using when estimating the population size or average density (including figures 3 and 4a–e)? Or is it ensemble of all models? This is nevertheless important information and if it is missing, you should clearly provide the information in the text and also to relevant figure captions.

7.

Mean temperature of the warmest month seems to be one of the most important limiting factors of hunter-gatherer density in Europe (Table 1, Figure 4). It is therefore interesting that its impact on maximum hunter-gatherer density is negative between 22kyBP and 8kyBP (figure 5B). The figure 1 shows that between 10 and 15 C the 90th quantile of MWM is decreasing, because of couple of outlier points. These kind of "edge effects" are a known problem in GAMs and therefore there are different kinds of constrained GAMs available:

https://www.researchgate.net/publication/271740857_Shape_constrained_additive_models

<https://arxiv.org/pdf/1812.07696.pdf>

It is nevertheless quite unrealistic to assume that increase of MWM would have had negative impact on forager density from the LGM to Mid Holocene and I therefore suggest that you either try to use more conservative smoothing parameter value to get rid of wiggles or switch to constrained GAM, although I don't know if there are quantile versions available for such techniques. The negative (but unrealistic) effect of MWM is at least one of the reasons why MWM appears to increase its importance as a limiting factor over time in Europe.

8.

On page 5, you describe your results so that during the LGM the northern limit of human range would have been in central France and southern Germany. However, my reading of figure 4 is that the whole of France would have been within the human range. You use one individual/100km² as threshold for human occupancy, which is pretty high given that lowest densities in ethnographic data are 0.2–0.25 individuals/100km². But even with your threshold, the occupied area seems to be clearly bigger than you describe in the text. Why this discrepancy? I would suggest that you bravely stand behind your results and describe them as they are :-)

Reviewer #3:

Remarks to the Author:

The manuscript focuses on the relation between the environmental factors explored here and population density. One central assumption that is adopted in this paper is that for foragers, demographic and environmental changes correlate strongly. And that there are causal relations between different environmental variables and human responses through time and Space. They then focus on limiting environmental factor which are defined as the variable predicting the lowest population density at a given place and time and assume that one of these limiting factors, or a combination of several, limited the scarcest resource, and in turn regulate population sizes and densities. They then identify the dominant climatic constraints for hunter-gatherer population densities and then hindcast their changing dynamics in Europe for the period between 20kyBP to 8kyBP. They detect spatiotemporal variations in these factors in relation to the assessed demographic data for human groups which suggests that European Upper Palaeolithic hunter-gatherers at various regions and periods needed to overcome very different adaptive challenges.

The paper is overall well written and the introduction and Results and Discussion are detailed, and cite a lot of relevant and up to date sources. Moreover, the main caveats associated with their data sources and analyses are mentioned and discussed

I would like to raise three issues which I think can be handled in the revised manuscript.

One is that while I agree that environmental changes seem to have been the main driving force behind evident demographic patterns in the case of human populations and various other species, as the authors indicate, there are also adaptive capacities of humans to buffer and manage at least to some extent, environmental changes and corresponding resource fluctuations. The cited paper by Filho et al. 2021, documents how several African communities differentially adapted to climate changes. If we assume that at least some of the human groups, during the Last Glacial Maximum and post-LGM period had similar adaptive capacities, it follows that their population sizes, densities and even settlement patterns, will not only reflect a 'passive' causal relationship with a specific climatic limiting factor, but also a unique human capacity to buffer and perhaps even overcome some limitation. Some examples include shelters, fire, and projectile technology. Moreover, one of the main mechanisms is mobility and mainly dispersals to refugia with better resources and climatic conditions.

A second issue is the reliance on ethnographic data. The authors cite the article by Bird a& Codding

2021, about the Promise and peril of ecological and evolutionary modelling using cross-cultural datasets. While the authors of this paper claim that the promise outweighs the peril. It will be useful for the authors to mention in more details, the potential caveats of drawing the analogy between present day and Upper Palaeolithic hunter-gatherers, since various papers argued that the former are not really a good proxy for the latter.

A third issue is that on page 6, Figure 3, they refer to the date of recolonization of Europe to be 17 kyBP. This is no longer regarded, on the basis of archaeological data, as being the date of onset of the process, as new results indicate that it started around 19 kyBP- see the paper by Maier et al. 2020: <https://doi.org/10.1007/s41982-019-00045-1>

In sum, the paper is informative and balanced but the above-mentioned points are raised as the way some of the text is worded, it seems that the underlying approach is that human demography is not only affected by environmental shifts, and more specifically climatic changes, but is directly caused only by these. In which case, the assessment of which specific limiting factor exerted the most impact on a given human populations at a given location and time is indeed informative and interesting. But it should be made clearer that the paper does not test the specific role of human cultural capacities, to buffer and even overcome some limiting factors. Moreover, the spatiotemporal variations are expected to be a reflection of the fact that indeed limiting factors varied and that hunter-gatherers needed to overcome different adaptive challenges, but they cannot shed light on how they actually adapted, or alternatively failed to adapt, to these changes.

Minor comments

Some of the figures need to be improved in terms of colors and legends.:

Figure 1. What are the abscissa? It is not clear from the figure legend.

Figure 3, Change color for Maximum temperature of the Warmest Month.

Figure 4, side panel legend, should be Population density and not population size
It is also difficult to understand the panels, what is the difference between each side and the colors are difficult to detect at this scale.

Figure 5, is it assessing population size or population density?

Final responses:

R1c1: Mean temperature of the Warmest Month (MWM) is identified a major limiting factor during all critical periods (Fig. 4), but MWM is the variable that has the most severe non-analogy problem comparing present-day climate space and past climates (Fig. 2 and Supplementary material S2). Although mentioned at Lines 126-128, considering the strong relevance to the main findings, the risk of an unreliable extrapolation out of the range of data used to fit the statistical model is higher than acknowledged here.

To acknowledge the points made by the reviewer regarding the non-analogy problem with the variable, Mean temperature of the Warmest Month, we remove this variable from the pool of factors used to assess population densities and limiting factors. We explain this in the text in L137-140 and L401-403. We do not consider that removing this variable is a significant problem for our analyses. We reason that our objective is not to determine how a specific variable determines population densities but on the possible processes (as we specify in Table 1) by which climate can determine population densities.

R1c2: It is not clear why the authors chose 90th percentile of population density to do the hindcast. First, the results, in principle, would not be comparable to previous estimates in literature. Second, does the resulted limiting factors change dependent on the choice of the percentile? Although the general shape of the population density versus climate relationship looks similar across different percentiles for each individual climate factor (Lines 311-312), it is the relative magnitudes between all predicted densities by these factors that ultimately selects the limiting factor. Thus, it is not straightforward whether your results are sensitive to the choice of percentiles.

The reviewer's point regarding the need to justify why we chose the 90th percentile in our analyses is welcomed. We have done all the analyses using the 10th, 50th, and 90th percentile in this revision. As we now clarify in the text (L62-63; L365-359; L407-412), our goal is not to quantify the population size on each evaluated grid but to indicate what are the potential climatic limiting factors and which could be the expected values (maximum/average/minimum) given this climatic limit. Furthermore, our results and discussion focus on how observed deviations from these estimates can be used to generate hypotheses to how different societies have (or not) tackled these climatic limits, allowing them to have larger population sizes.

R1c3: Regarding the comparison between hindcasted population density and the archaeological population proxy (Fig. 3), I would not say they are "in line with" each other (Lines 139-140). The black curve in Fig. 3 starts to increase already since 18 ka, which is relatively flat in the red curve; the red curve increases significantly during GI1 and GS1, whereas it is stale in the black curve.

While we now acknowledge the nuanced description of the trends by the reviewer in the text (L172-176), we consider that a perfect match on the timing of events cannot be expected as these are variables representing trends at two different resolutions. Having

said that, the archaeological population proxy and our population density estimates show a strong correlation ($\rho = -0.7$) when aggregated at the same temporal resolution as the archaeological population proxy. We now make this point explicit in our text (L172-176).

R1c4: By the way, at Lines 101-103, why do you separate higher and lower predictive accuracy by a threshold of “explained deviances < 0.79 ”? According to Table 1, these predictors are all so close.

We do make this distinction anymore. Now we acknowledge that there are differences in the predictive accuracy between variables, but that accuracy amongst predictors is somewhat similar (L122-124).

R1c5: From the results it is hard to infer mechanisms regarding how the identified limiting factor has constrained population density. This is limited by the fact that only temperature and precipitation and their variants were used as predictors, without direct information about productivity; whereas climate impacts population density via indirect effects on ecosystem attributes like NPP (e.g. Freeman et al. 2020, doi:10.1016/j.jas.2020.105168). Throughout the text the authors have tried to relate some of the factors to environmental productivity, but it was highly speculative. Let me take Lines 185-192 as an example. During 14.7ka to 11.7ka, the importance of ET decreases while importance of MWM and temperature seasonality increases. But all three variables are linked to NPP (and possibly other aspects of the ecosystem). From these changes one still cannot judge what process is taking effect in the end. Given this, why not use NPP as a predictor in the first place? Data availability for the hindcast should not be a problem since simulated NPP for the past 21,000 years are publicly accessible from some climate models already.

To address the comment, we have done two things:

First, we no use Net Primary Productivity (NPP) in our work as a predictor. Using the Miami model, we calculate this variable (Lieth, 1972, as described in Table 1). We use this modelling approach instead of other possible NPP products as we want to reduce the potential biases that could come from using environmental datasets from alternative sources. As we do this, NPP as a predictor shows that it is not a significant factor.

Second, we now refer to Effective Temperature and Potential Evapotranspiration as factors determining energy availability in a broad context (Table 1). NPP relates only to a variable indicating the energy available to hunter-gatherers from primary producers.

R1c6: I commend the authors’ effort to put the (more of ecology-oriented) results into archaeological context, but currently it is still limited in qualitative descriptions scattered in the text. If the authors could achieve a more systematic compilation of archaeological records regarding how these societies have tackled with the limiting climate factors and find a consistency with your hindcasted results in space and time, it would add much merit to this study, with broader significance and impacts.

Thank you for this comment – naturally, we love to expand on this particular issue. We now provide an extended discussion of how the archaeological record explicitly links to the identified limiting factors and how different forager groups overcame these. We also provide additional references relating to pyrotechnology, shelter, energy capture, etc. (e.g. . L223-226 and L262-271).

R1c7: Aside from the above concerns, the organization of Results and discussion needs to be improved. Adding sub-headings would help. Besides, descriptions of the results in the text should be more careful. Currently they are sometimes inconsistent with the table or figures. For example, at Lines 104-110, it says seasonal temperature variables are among the lowest explained deviance, which is not the case as listed in Table 1.

As suggested, we have added subheadings to the Results and discussion section to provide a clear outline of our study results and their implication and relevance. We have now addressed all inconsistencies between the tables, figures and text.

R1c8: A minor point is that the uncertainties/biases in the paleoclimate outputs of the CCSM3 climate model should be discussed.

We would like to evaluate and discuss how CCSM3 SynTrace paleoclimate simulations uncertainties propagate to our population density models and definition of limiting factors. However, the used downscaled and debiased paleoclimatic simulations do not contain uncertainty estimates, and this is a point we acknowledge in our manuscript methods (L394-397).

R1c9: Code availability: though not mandatory, it is strongly encouraged to make the code readily available so as to enable reproduction of the results.

We have now made the code and data used in this study available through a project GitHub site: <https://github.com/AlejoOrdonez/PaleoPopDen>. This is now part of the data availability statement.

R1c10: Table 1: A conceptual confusion: MCM is not “extreme events”. Same for MWM, PDM, and PWM. Extreme events are events that occur with low frequency, not the regular seasonal maxima or minima.

We have now renamed these as annual limits.

R1c11: Line 266: “16 climatic predictors”: there are 18 climate variables in Table 1.

We have now change these to the current number of predictors.

R1c12: Line 289: it says “We use a subsample of 159 hunter-gatherers populations...” in the Reporting summary, while here it says “127 populations”.

We have corrected this to there is consistency with the **Reporting summary**.

R1c13: Table 1: Acronym of “Precipitation of the Wettest Month” should be PWM, not PDM.

We have corrected this as suggested.

R1c14: Figure caption of Fig. 3: “Minimum temperature of the Coldest Month” should be “Mean...”, and “Maximum temperature of the Warmest Month” should be “Mean...”
We have changed the figure layout to include all used variables and ensure the titles and legend match Table 1

R1c15: Figure caption of Supplementary material S1: it is not “the six most important environmental factors”.
We have removed this figure as all regressions are now shown in the main text.

R2c1. Given that many of the climate predictors are highly correlated (as you also write in the manuscript), they will convey partly the same information. This can potentially make it difficult to differentiate the importance between different climatic variables as limiting factors. I would suggest that you add some more justification for using large number correlated variables in the analysis. Or, alternatively, consider reducing the dimensionality of the data.

While we acknowledge in our study that the level of correlation between predictors is high, this level of relationship amongst predictors allows us to “[justify] our grouping of individual variables within groups of possible explanatory mechanisms (as listed in **Table 1**)” (L69-74; L124-129; and L326-327). We also provide further justifications for this in our response to the reviewer’s flowing point.

R2c2. Partly related to the item 1, you could provide justification for using univariate instead of multivariate approach. The effect of a predictor variable can change (sometimes dramatically) when controlled for the effects of other variables by adding them to the model. Therefore, you should explain why you rely on univariate approach, or, alternatively, try to add the best predictor candidates in the same model and see how the results would change.

As we now explicitly state in our text, “*we are not aiming at determining the best combination of variables to predict population density, but rather at determining the limiting effect of a given environmental driver*” (L69-74). This perspective aligns with the core idea of limiting factors behind the current study.

R2c3. Binford’s data is notorious for spatial auto-correlation, especially because in particular areas, he has basically split closely living ecologically, demographically and culturally similar groups into smaller units even though one might consider many of those belonging to the same ethnic group. This can lead to inflated performance metrics in traditional cross-validation schemes. The idea of cross-validation is to test the model with data that the model has not seen before, so in the presence of spatial auto-correlation, test data can be “too” similar to training data. Therefore, your performance metrics are quite likely “too good” and I suggest that you could use some kind of spatial block cross-validation scheme, such as h-block cross validation.

As suggested, we have used an h-block cross-validation approach (L68; L377-383; L398-401) to determine, for each qGAM model, its' performance and use these multiple models to control for model specification variability in our estimates of Population Density.

R2c4. Related to the above issue, on page 3 you write that no single environmental variable explained more than 81% of the population density variation among ethnographic foraging societies. It might be because of my ignorance of quantile regression, but I'm not sure if you can really say that quantile regression model can explain some percentage of the variation in a response variable. So clarify this and explain what the explained deviance is measuring in your quantile regressions, is it the goodness of fit of the 90th quantile or what?

You are right in your assessment that quantile GAMs cannot provide an estimate of the "percentage of the variation in a response variable" (i.e., R^2). Our values here refer to the 50th percentile qGAM (or a traditional GAM), a point that was not clear in the original submission. For these, it is possible to determine an R^2 value. This revision ensures that the point is explicitly made in the text (L374-377). In both the main text and the method section (L110-120; Table 1), we now also describe the model deviance.

R2c5. I'm also not sure if one can directly compare your performance metrics (explained deviance) to e.g. our metrics (R^2) (Tallavaara et al. 2015). Besides, our R^2 e.g. for multivariate GAM is clearly smaller (0.6), not marginally better, than any your values. After a lot of exploration with Binford's data, I also think that it is really difficult to push the R^2 of (multivariate) population density models well above 0.7 unless you really overfit the model.

We agree that a proper comparison between the performance of our models and those in other publications is not so straightforward. Therefore decided to omit this statement in the revised text.

R2c6. On page 5, you provide the modelled population size estimates for Europe, which seems to be pretty high. The LGM estimate is twice as large as our previous estimate (which has been argued to be way too large by some) despite we having larger geographical area. However, am I right that your estimates are actually maximum estimates based on the modelled 90th quantile? Whatever the case, this needs to be stated clearly in the text and in the relevant figure captions.

We are aware of this, and it is a result of us using the 90th percentile model when describing these trends – as accurately pointed out in the comments. This revision states that "*Taken at face value, these figures are gross overestimations of actual sustained and demographically viable human land-use across this timeframe*" (L169-170). Furthermore, we state in the text that our goal is NOT to predict population density but rather to show the limiting effects of climate on this important variable (L74-76). Therefore, it makes sense to consider maximum (90th-percentile), average (50th-percentile), and minimum (10th-percentile) values as descriptors of these possible

limits. These are clarifications we also make when describing our population size/density estimates (L154-169; L177-180).

R2c7. I might have missed it somehow, but which of the many univariate models you are using when estimating the population size or average density (including figures 3 and 4a–e)? Or is it ensemble of all models? This is nevertheless important information and if it is missing, you should clearly provide the information in the text and also to relevant figure captions.

This information was only in the methods in the original submission (L386-397), and now it is part of the main text (L84-89) and the relevant legends.

R2c8. Mean temperature of the warmest month seems to be one of the most important limiting factors of hunter-gatherer density in Europe (Table 1, Figure 4). It is therefore interesting that its impact on maximum hunter-gatherer density is negative between 22kyBP and 8kyBP (figure 5B). The figure 1 shows that between 10 and 15 C the 90th quantile of MWM is decreasing, because of couple of outlier points. These kind of “edge effects” are a known problem in GAMs and therefore there are different kinds of constrained GAMs available.

It is nevertheless quite unrealistic to assume that increase of MWM would have had negative impact on forager density from the LGM to Mid Holocene and I therefore suggest that you either try to use more conservative smoothing parameter value to get rid of wiggles or switch to constrained GAM, although I dont know if there are quantile versions available for such techniques. The negative (but unrealistic) effect of MWM is at least one of the reasons why MWM appears to increase its importance as a limiting factor over time in Europe.

Thanks for your point regarding the patterns in this variable. This is one of the points we have been discussing in our revision. Given the issues highlighted in this comment and the points raised by Reviewer-1 (the fact that there is a large non-analogy for this variable, especially in the late Pleistocene), we have decided to remove this variable from our analyses. As we discussed in R1C1, we do not consider this a significant problem for our work. Our reasoning is that because our focus is mainly on the “environmental mechanisms” by which climate imposes a limitation to population density (captured usually by two to three variables in our dataset) and not the effect of an individual variable.

R2c9. On page 5, you describe your results so that during the LGM the northern limit of human range would have been in central France and southern Germany. However, my reading of figure 4 is that the whole of France would have been within the human range. You use one individual/100km² as threshold for human occupancy, which is pretty high given that lowest densities in ethnographic data are 0.2–0.25 individuals/100km². But even with your threshold, the occupied area seems to be clearly bigger than you describe in the text. Why this discrepancy? I would suggest that you bravely stand behind your results and describe them as they are.

This text section is now modified (L188-192) to reflect a more detailed discussion of the observed pattern.

R3c1. One is that while I agree that environmental changes seem to have been the main driving force behind evident demographic patterns in the case of human populations and various other species, as the authors indicate, there are also adaptive capacities of humans to buffer and manage at least to some extent, environmental changes and corresponding resource fluctuations. The cited paper by Filho et al. 2021, documents how several African communities differentially adapted to climate changes. If we assume that at least some of the human groups, during the Last Glacial Maximum and post-LGM period had similar adaptive capacities, it follows that their population sizes, densities an even settlement patterns, will not only reflect a ‘passive’ causal relationship with a specific climatic limiting factor, but also a unique human capacity to buffer and perhaps even overcome some limitation. Some examples include shelters, fire, and projectile technology. Moreover, one of the main mechanisms is mobility and mainly dispersals to refugia with better resources and climatic conditions.

The reviewer points to one of the main points we wanted to showcase with this study, but perhaps it was not clear – that climate sets a stage for human adaptation to “act” (L289-297). You could see this as climate determining a baseline “limit”, where human-populations active interaction with the environment, via behaviour and tools, would result in a deviation from this limit. This is a point we make explicit in our text (L293-297), indicating that deviations from our estimates can be used to signpost which populations had buffering strategies and generate hypotheses as to which could these strategies be.

R3c2. A second issue is the reliance on ethnographic data. The authors cite the article by Bird a& Codding 2021, about the Promise and peril of ecological and evolutionary modelling using cross-cultural datasets. While the authors of this paper claim that the promise outweighs the peril. It will be useful for the authors to mention in more details, the potential caveats of drawing the analogy between present day and Upper Palaeolithic hunter-gatherers, since various papers argued that the former are not really a good proxy for the latter.

A paragraph on the inferential limits of the available ethnographic datasets has been added (L148-152). However, we do consider a very detailed discussion of these issues outside of the scope of this particular study, not least because it has been discussed directly in the recent literature, e.g.: Hamilton, M.J., Tallavaara, M., 2022. Statistical inference, scale and noise in comparative anthropology. *Nature Ecology & Evolution* 6, 122–122. <https://doi.org/10.1038/s41559-021-01637-3>

R3c3. A third issue is that on page 6, Figure 3, they refer to the date of recolonization of Europe to be 17 kyBP. This is no longer regarded, on the basis of archaeological data, as being the date of onset of the process, as new results indicate that it started

around 19 kyBP- see the paper by Maier et al.

2020: <https://doi.org/10.1007/s41982-019-00045-1>

This text section has been amended (**L201-203**) and the appropriate reference added.

R3c4. In sum, the paper is informative and balanced but the above-mentioned points are raised as the way some of the text is worded, it seems that the underlying approach is that human demography is not only affected by environmental shifts, and more specifically climatic changes, but is directly caused only by these. In which case, the assessment of which specific limiting factor exerted the most impact on a given human populations at a given location and time is indeed informative and interesting. But it should be made clearer that the paper does not test the specific role of human cultural capacities, to buffer and even overcome some limiting factors. Moreover, the spatio-temporal variations are expected to be a reflection of the fact that indeed limiting factors varied and that hunter-gatherers needed to overcome different adaptive challenges, but they cannot shed light on how they actually adapted, or alternatively failed to adapt, to these changes.

Thank you for the thoughtful summary of our ideas in our study. We have now added text to ensure the points the reviewer so correctly highlights are even more evident in the text. Notably, the ideas of environmental conditions as factors affecting and determining human demography in the evaluated period (L42-44) determine how technology or behaviour resulted in particular populations overcoming the limitations imposed by the rapid climatic changes of the late-Pleistocene (L74-78).

R3c5. Some of the figures need to be improved in terms of colors and legends.:

We have done a substitution change in the figures color and legend to clarify their message, and fully explain what the objective of these is.

R3c6. Figure 1. What are the abscissa? It is not clear from the figure legend.

Figure 1 now show what is the variable in the Abscissa (the same as the title)

R3c7. Figure 3, Change color for Maximum temperature of the Warmest Month.

In figure-1, we have now plotted all the used variables and used a colour scheme that facilitates the readability of the variables.

R3c8. Figure 4, side panel legend, should be Population density and not population size. It is also difficult to understand the panels, what is the difference between each side and the colors are difficult to detect at this scale.

Figure one has been redrawn, and the density and limiting factors maps have been separated to clarify and enhance the message of each plot.

R3c9. Figure 5, is it assessing population size or population density?

We now clarify that the top panel shows the changes in the evaluated period (21kyBP to 8kyBP) in the proportion of ice-free cells where a viable is considered the limiting

factor (predicts the min population density). The bottom panel shows the estimated population density based on the average climatic condition across Europe for each evaluated variable.

Reviewers' Comments:

Reviewer #1:

Remarks to the Author:

The authors have addressed my major comments by 1) removing the MWM variable in assessing the limiting climate factors so that the serious non-analogy problem can be bypassed; 2) testing NPP as a potential limiting factor in the analysis; and 3) extending the discussion regarding the significance of the results in archaeological perspective. Overall I'm satisfied with these revisions. But there are some points to be clarified in the revised manuscript:

For the variable temperature seasonality, why are the values so large, ~2000 °C (Figure 1F and Figure 2F)? How is it calculated? And for precipitation seasonality, is it the standard deviation of the monthly precipitation?

Table 1: why are the metrics substantially lower than that in the previous version? Is it because now you have used h-block cross validation to address spatial auto-correlation? Besides, why do the deviance explained and R^2 differ so much for some variables like PWM and TAP?

Line 165: what is the threshold of population density to define an occupied grid cell?

Figure 4: It would be more informative if you could overlay the localities of the archaeological sites that correspond to each time interval on the predicted population density maps. It can serve as a qualitative comparison. Besides, the current color legend looks weird – the ticks are not at the boundaries of each color segment.

In addition, there are still quite a few careless errors and inconsistencies in the revised manuscript. Below are some examples.

Line 110: "most of environmental variable produced models that explaining over 50% of the population density variation" – according to Table 1, the explanatory power of the variables are mostly below 50%.

Line 230: "PET" here should be TAP?

Line 234: "PET and TAP were the main limiting factors" – according to Fig. 6, it should be TS and TAP.

Line 254-255: MWM is no longer used in the prediction.

Table 1: "TSeson" and "PREC" – inconsistent with those in the text. And check the footnote of Table 1.

Figure 1: The precipitation has been log-transformed, right? Need to specify it.

Figure 3 lower panel: why Precip. Driest Month is higher than Precip. Wettest Month?

Figure 5 caption: what is "F-J"?

It is the authors' job to closely check every sentence, figures and tables to avoid any inconsistency or contradiction!

Reviewer #2:

Remarks to the Author:

Authors have successfully revised their manuscript. I have just one follow-up comment because authors might have misunderstood my earlier comment about models having one predictor at a time. My intention was not to suggest to add multiple predictors to achieve better predictive ability, but to take into account the fact that the effect of a predictor can change when one takes into account the effect(s) of other potential predictor(s).

For example, ET and MCM both appear to be important limiting factors and also representing different kinds of limiting factors, ET relating to energy availability and MCM to annual limits. However, these variables are also highly correlated, which already indicates that it will be difficult to tell apart their

individual effects. When you include both variables as predictors in the same model it actually turns out that the effect of ET is not statistically significant, response of population density to ET being more or less flat. Similarly, if you add e.g. NPP, ET and TS to the same model their effect (response shapes) are different from their effect when each is the only predictor in the model.

To me, all this suggests that the real limiting effects of climate variables can be different from those you get when you include these variables separately as predictors. However, I don't know how severe issue this truly is, but I would like to know your thoughts on that. If it really is an issue, one might use PCA to create uncorrelated climate variables and use these as predictors in the models.

Best wishes,
Miikka Tallavaara

Reviewer #3:

Remarks to the Author:

I am fully satisfied with the revised version and with the revised manuscript and the changes.

REVIEWER COMMENTS

Reviewer #1

R12Co. The authors have addressed my major comments by 1) removing the MWM variable in assessing the limiting climate factors so that the serious non-analogy problem can be bypassed; 2) testing NPP as a potential limiting factor in the analysis; and 3) extending the discussion regarding the significance of the results in archaeological perspective. Overall, I'm satisfied with these revisions. But there are some points to be clarified in the revised manuscript:

We appreciate your assessment regarding our revision.

R1C1. For the variable temperature seasonality, why are the values so large, ~2000 °C (Figure 1F and Figure 2F)? How is it calculated?

Thanks for bringing this to our attention. Temperature Seasonality (TS) is estimated as the SD of mean annual temperatures X 100. For Clarity, we have now done two things. First, we now clarify that TS is measured as the SD of mean annual temperatures (so that values are in the same order of magnitude as other temperature variables). Second, we specify how (TS) is calculated in Table 1.

R1C2. And for precipitation seasonality, is it the standard deviation of the monthly precipitation?

As for TS, we now explain in table 1 how precipitation seasonality (PS) is estimated. In short, yes, it is calculated as the variation in monthly precipitation. However, instead of the SD in monthly precipitation, we use the Coefficient of Variation (CV) as this is the standard when estimating bioclimatic variables. We also clarify this in the "variable" and "units" columns of Table 1.

R1C3. Table 1: why are the metrics substantially lower than that in the previous version? Is it because now you have used h-block cross validation to address spatial autocorrelation? Besides, why do the deviance explained and R^2 differ so much for some variables like PWM and TAP?

Yes, the values are lower due to using an h-block cross-validation approach to define the random samples. Furthermore, two points explain the lower deviance-explained when compared to the R^2 values. First, adding the variable does not add more explanatory power to the model compared to an intercept-only model (*hence the low deviance explained and likely low unadjusted R^2*). Second, we can interpret the higher R^2 as the models built on the training dataset can accurately describe the test dataset, which ensures the idea of model transferability. To ensure these points are clear, we add these points of clarification to table 1 legend.

R1C4. Line 165: what is the threshold of population density to define an occupied grid cell?

Ecoinformatics and Biodiversity

Alejandro Ordonez Gloria
Assistant Professor

Date: 6 July 2022

Direct Tel.: +45 8715 4347
Mobile Tel.: +4550229006
E-mail: alejandro.ordonez@bio.au.dk
Web: au.dk/en/alejandro.ordonez@bio

Sender's CVR no.:
31119103

Page 1/4

A cell was defined as occupied if our model predicted population densities above 0.2 individuals per 100km² (the lowest densities in the ethnographic dataset). This point is added to the main text (L162-163) and the methods (L405-406).

R1C5. Figure 4: It would be more informative if you could overlay the localities of the archaeological sites that correspond to each time interval on the predicted population density maps. It can serve as a qualitative comparison. Besides, the current color legend looks weird – the ticks are not at the boundaries of each color segment.

We explored adding the localities of the archaeological sites to figure 4 but decided not to include these as these create an unnecessary layer of complexity for the figure. We also address the point raised by the reviewer regarding the figure colour legend.

R1C6. Line 110: “most of environmental variable produced models that explaining over 50% of the population density variation” – according to Table 1, the explanatory power of the variables are mostly below 50%.

We appreciate the reviewer catching this inconsistency coming from a legacy text from the first version. We now changed the sentence, so it does not specify a cut-off value (50%) but the range of mean deviance across the 1000 different models (L110).

R1C7. Line 230: “PET” here should be TAP?

We consider that here the variable to include is PET, as we are building from the idea of a relationship between productivity and Evapotranspiration. This is the case as the second point relates to energy availability, not climate variability. To further justify this link, we add a reference (L232).

R1C8. Line 234: “PET and TAP were the main limiting factors” – according to Fig. 6, it should be TS and TAP.

We appreciate the reviewer catching this inconsistency coming from a legacy text from the original submission. We now changed the sentence accordingly.

R1C9. Line 254-255: MWM is no longer used in the prediction.

We appreciate the reviewer catching this legacy text from the first submission. We have rephased this point (L256).

R1C10. Table 1: “TSeson” and “PREC” – inconsistent with those in the text. And check the footnote of Table 1.

The acronym was changed as suggested in the table for consistency with the text.

R1C11. Figure 1: The precipitation has been log-transformed, right? Need to specify it.

We have added this clarification to the corresponding axes in figure 1 and table 1.

R1C12. Figure 3 lower panel: why Precip. Driest Month is higher than Precip. Wettest Month?

We appreciate the reviewer catching this inconsistency. This was a problem in the code calling the different variables after we removed the Temperature of the Warmest Month, which caused a mismatch between the names and the data plotted in the bottom panels of figure 3. The figure now has corrected this.

R1C13. Figure 5 caption: what is “F-J”?

We appreciate the reviewer catching this legacy text from the first submission. This text was removed.

Reviewer #2.

R2C1. Authors have successfully revised their manuscript. I have just one follow-up comment because authors might have misunderstood my earlier comment about models having one predictor at a time.

My intention was not to suggest to add multiple predictors to achieve better predictive ability, but to take into account the fact that the effect of a predictor can change when one takes into account the effect(s) of other potential predictor(s).

For example, ET and MCM both appear to be important limiting factors and also representing different kinds of limiting factors, ET relating to energy availability and MCM to annual limits. However, these variables are also highly correlated, which already indicates that it will be difficult to tell apart their individual effects. When you include both variables as predictors in the same model it actually turns out that the effect of ET is not statistically significant, response of population density to ET being more or less flat. Similarly, if you add e.g. NPP, ET and TS to the same model their effect (response shapes) are different from their effect when each is the only predictor in the model.

To me, all this suggests that the real limiting effects of climate variables can be different from those you get when you include these variables separately as predictors.

However, I don't know how severe issue this truly is, but I would like to know your thoughts on that. If it really is an issue, one might use PCA to create uncorrelated climate variables and use these as predictors in the models.

We thank the reviewer for his positive feedback on our revision and the clarification of his original point. As we now understand the reviewer's point, the issue is that significant absolute effects from univariate models would not translate into relative effects determined by multivariate models.

While we agree with the point, we consider that the multiple regression approach does not translate to the idea of limiting factors we are evaluating here. We argue that multiple regression coefficients indicate effects in the context of other variables (hence contingent on which variables are included or omitted in a model). Therefore, these determine how much each variable contributes to the change in population density. To define which variables set a lower boundary, we require a measure of absolute effects provided by univariate approaches. Focusing on the relative effects would not allow us

to define limiting factors but which variable(s) contribute the most to changes in population density from the "regional" mean.

Suppose we could build models for change in population density for each evaluated time bin. In that case, we could define the variable that contributes the most to population density at each time bin. Still, this is not a limiting factor but the variable that contributes the most to changes in population density from the regional average (i.e., the model intercept). Last, there is the issue of translating these relative effects into space, which our approach based on univariate models can do.

Furthermore, while PCA, or other ordination approaches, could be used here to determine "groups of variables" and the variable most representative of such "group", we will still be looking at relative effects when using the two or three most important axes.

There also be questions about how suitable it is to use the eigenvectors generated by the ordination under current conditions to "reorganize" past climatic surfaces where the correlations between variables change.

In summary, we consider that using univariate models, while far from perfect, is a practical approach to assessing the absolute effects of each variable and comparing these between variables over time. Also, it allows us to link our models to a process. All these points are now made in the text (L72-80) and the methods (L364-368).

Reviewer #3.

I am fully satisfied with the revised version and with the revised manuscript and the changes.

Thanks for your positive assessment.

Reviewers' Comments:

Reviewer #1:

Remarks to the Author:

I'm satisfied with the revisions and have no further comments.

Reviewer #2:

Remarks to the Author:

While I still slightly disagree with you about the effects of univariate vs multivariate models on the results, I'm happy to do so. It is good that you now explain in the manuscript your choices regarding the matter, so I'm fully satisfied with this revised manuscript.

Best,

Miikka Tallavaara

REVIEWER COMMENTS

Page 2/2

Reviewer #2 (Remarks to the Author):

While I still slightly disagree with you about the effects of univariate vs multivariate models on the results, I'm happy to do so. It is good that you now explain in the manuscript your choices regarding the matter, so I'm fully satisfied with this revised manuscript.

We appreciate your assessment regarding our revision and your willingness to agree to disagree regarding the effects of univariate vs multivariate models on the results.